# Bandit Phase Retrieval

**Tor Lattimore**
DeepMind, London
lattimore@deepmind.com

**Botao Hao**
DeepMind, London
haobotao000@gmail.com

## Abstract

We study a bandit version of phase retrieval where the learner chooses actions $(A_t)_{t=1}^n$ in the $d$-dimensional unit ball and the expected reward is $\langle A_t, \theta_\star \rangle^2$ with $\theta_\star \in \mathbb{R}^d$ an unknown parameter vector. We prove an upper bound on the minimax cumulative regret in this problem of $\tilde{\Theta}(d\sqrt{n})$, which matches known lower bounds up to logarithmic factors and improves on the best known upper bound by a factor of $\sqrt{d}$. We also show that the minimax simple regret is $\tilde{\Theta}(d/\sqrt{n})$ and that this is only achievable by an adaptive algorithm. Our analysis shows that an apparently convincing heuristic for guessing lower bounds can be misleading and that uniform bounds on the information ratio for information-directed sampling [Russo and Van Roy, 2014] are not sufficient for optimal regret.

## 1 Introduction

We study an instantiation of the low-rank bandit problem [Jun et al., 2019] that in the statistical setting is called phase retrieval. Although this model is interesting in its own right, our main focus is on the curious information structure of this problem and how it impacts algorithm design choices. Notably, we were not able to prove optimal regret for standard approaches based on optimism, Thompson sampling or even information-directed sampling. Instead, our algorithm is a variant of explore-then-commit with an adaptive exploration phase that learns to gain information at a faster rate than what is achievable with non-adaptive exploration.

**Problem setting** Let $\| \cdot \|$ be the standard euclidean norm and $\mathbb{B}_r^d = \{x \in \mathbb{R}^d : \|x\| \le r\}$ and $\mathbb{S}_r^{d-1} = \{x \in \mathbb{R}^d : \|x\| = r\}$. At the start of the game the environment secretly chooses a vector $\theta_\star \in \mathbb{S}_r^{d-1}$ with $r \in [0,1]$ a constant that is known to the learner. The assumption that $r$ is known can be relaxed at essentially no cost (Section 8). The game then proceeds over $n$ rounds. In round $t$ the learner chooses an action $A_t \in \mathbb{B}_1^d$ and observes a reward

$$X_t = \langle A_t, \theta_\star \rangle^2 + \eta_t \,,$$

where $(\eta_t)_{t=1}^n$ is a sequence of independent standard Gaussian random variables. As is standard in bandit problems, the conditional law of $A_t$ should be chosen as a (measurable) function of the previous actions $(A_s)_{s=1}^{t-1}$ and rewards $(X_s)_{s=1}^{t-1}$ and possibly an exogenous source of randomness. The performance of a policy $\pi$ is measured in terms of the expected regret,

$$\mathfrak{R}_n(\pi, \theta_\star) = \max_{a \in \mathbb{B}_1^d} \mathbb{E}\left[\sum_{t=1}^n \left(\langle a, \theta_\star \rangle^2 - \langle A_t, \theta_\star \rangle^2\right)\right] = nr^2 - \mathbb{E}\left[\sum_{t=1}^n \langle A_t, \theta_\star \rangle^2\right] \,.$$

The minimax regret is $\mathfrak{R}_n^\star = \sup_{r \in [0,1]} \inf_\pi \sup_{\theta_\star \in \mathbb{S}_r^{d-1}} \mathfrak{R}_n(\pi, \theta_\star)$, where the infimum is over all policies.

We also study the pure exploration setting, where at the end of the game the learner uses the observed data $(A_t)_{t=1}^n$ and $(X_t)_{t=1}^n$ to make a prediction $\widehat{A}_\star \in \mathbb{B}_1^d$ of the optimal action. The simple regret of

policy $\pi$ is

$$\mathfrak{r}_n(\pi, \theta_\star) = \max_{a \in \mathbb{B}_1^d} \mathbb{E}\left[\langle a, \theta_\star \rangle^2 - \langle \widehat{A}_\star, \theta_\star \rangle^2\right] = r^2 - \mathbb{E}\left[\langle \widehat{A}_\star, \theta_\star \rangle^2\right].$$

As expected, the minimax simple regret is $\mathfrak{r}_n^\star = \sup_{r \in [0,1]} \inf_\pi \sup_{\theta_\star \in \mathbb{S}_r^{d-1}} \mathfrak{r}_n(\pi, \theta_\star)$.

**Contributions** Our main contribution is an upper bound on $\mathfrak{R}_n^\star$ that matches existing lower bounds up to logarithmic factors. For the simple regret we provide a near-optimal upper bound and a lower bound showing that non-adaptive policies must be at least a factor of $\Omega(\sqrt{d})$ suboptimal. In all of the following, const is a universal non-negative constant that may vary from one expression to the next.

**Theorem 1.** $\mathfrak{R}_n^\star \leq \text{const } d\sqrt{n \log(n) \log(d)}$.

A corollary of the proof of the lower bound by Kotłowski and Neu [2019] shows that the minimax regret is at least $\Omega(d\sqrt{n})$, so the minimax regret for bandit phase retrieval is now known up to logarithmic factors. For the simple regret we provide the following upper and lower bounds:

**Theorem 2.** $\mathfrak{r}_n^\star \leq \text{const } d\sqrt{\log(n) \log(d)/n}$.

**Theorem 3.** *Assume that $n \geq d \geq 8$. Then there exists an $r \in [0,1]$ such that for all policies $\pi$ with $(A_t)_{t=1}^n$ independent of $(X_t)_{t=1}^n$, $\sup_{\theta_\star \in \mathbb{S}_r^{d-1}} \mathfrak{r}_n(\pi, \theta_\star) \geq \text{const } \sqrt{d^3/n}$.*

We also show that worst-case bounds on the information ratio for information-directed sampling are *not* sufficient to achieve optimal regret. Our results suggest that the conjectured lower bounds for low-rank bandits [Jun et al., 2019, Lu et al., 2021] are not true and that existing upper bounds may be loose. The same phenomenon may explain the gap between upper and lower bounds for bandit principle component analysis [Kotłowski and Neu, 2019], as we discuss in Section 8.

**Notation** The first $n$ integers are $[n] = \{1, 2, \ldots, n\}$ and the standard basis vectors in $\mathbb{R}^d$ are $e_1, \ldots, e_d$. The span of a collection of vectors is denoted by $\text{span}(v_1, \ldots, v_m)$ and the orthogonal complement of a linear subspace $V \subset \mathbb{R}^d$ is $V^\perp = \{x \in \mathbb{R}^d : \langle x, y \rangle = 0 \text{ for all } y \in V\}$. The mutual information between random elements $X$ and $Y$ on the same probability space is $I(X; Y)$ and the relative entropy between probability measures $P$ and $Q$ on the same measurable space is $\text{KL}(P, Q)$. The dimension of a set $\Theta \subset \mathbb{R}^d$ is defined as the dimension of the affine hull of $\Theta$.

## 2 Related work

The most related work is by Huang et al. [2021]. As happens surprisingly often, this is a case where multiple groups tackled the same problem and developed similar methods to arrive at approximately the same solution. Their work provides the same upper bound in the cumulative regret setting as we do using an explore-then-commit principle and the noisy power method. They also generalise the setting to consider higher powers. For example, when the reward is $\langle A_t, \theta_\star \rangle^3$. What is different is that we use a more ad-hoc (but still interesting) construction during the exploration period and our work is more focussed on the information-theoretic implications.

**Phase retrieval** Phase retrieval is a classical problem in signal processing and statistics [Candès et al., 2015b,a, Cai et al., 2016, Chen and Candès, 2017, Chen et al., 2019, Sun et al., 2018]. These works are focused on learning $\theta_\star$ where the covariates $(A_t)_{t=1}^n$ are uncontrolled, either random or fixed design.

**Linear bandits** Our problem can be written as a stochastic linear bandit by noticing that $\langle A_t, \theta_\star \rangle^2 = \langle A_t A_t^\top, \theta_\star \theta_\star^\top \rangle$, where the inner product between matrices on the right-hand side should be interpreted coordinate-wise and the action set is $\{aa^\top : a \in \mathbb{B}_1^d\}$. There is an enormous literature on stochastic linear bandits [Auer, 2002, Dani et al., 2008, Rusmevichientong and Tsitsiklis, 2010b, Chu et al., 2011, Abbasi-Yadkori et al., 2011]. This reduction immediately yields an upper bound on the minimax regret of $O(d^2\sqrt{n} \log(n))$.

**Low-rank bandits** Low-rank bandits are a kind of linear bandit where the environment is determined by an unknown matrix and the actions of the learner are also matrices. Let $\mathcal{E} \subset \mathbb{R}^{d_1 \times d_2}$ and $\mathcal{A} \subset \mathbb{R}^{d_1 \times d_2}$. A low-rank bandit problem over $\mathcal{E}$ and with actions $\mathcal{A}$ is characterised by a matrix $\Theta_\star \in \mathcal{E}$. The learner plays actions $A_t \in \mathcal{A}$ and the reward is $X_t = \langle A_t, \Theta_\star \rangle + \eta_t$, where $\eta_t$ is noise and the inner product between matrices is interpreted coordinate-wise. So far this is nothing more than

**Table 1:** For the low-rank bandits column, $p$ is the rank. We ignore logarithmic factors and universal constant. Note, the $dp\sqrt{n}$ lower bound derived by Lu et al. [2021] does not apply to bandit phase retrieval because it makes use of the richer structure of the more general model.

| Upper bounds | Bandit phase retrieval | Low-rank bandits | Pure exploration |
|---|---|---|---|
| Abbasi-Yadkori et al. [2011] | $O(d^2\sqrt{n})$ | $O(d^2\sqrt{n})$ | N/A |
| Jun et al. [2019], Lu et al. [2021] | $O(d^{3/2}\sqrt{n})$ | $O(d^{3/2}\sqrt{pn})$ | N/A |
| This work | $O(d\sqrt{n})$ | N/A | $O(d/\sqrt{n})$ |
| **Lower bounds** | | | |
| Lu et al. [2021] | N/A | $\Omega(dp\sqrt{n})$ | N/A |
| Kotłowski and Neu [2019] | $\Omega(d\sqrt{n})$ | N/A | N/A |
| This work | N/A | $\Omega(d\sqrt{n})$ | $\Omega(d/\sqrt{n})$ |
| This work (non-adaptive learning) | N/A | N/A | $\Omega(d^{3/2}/\sqrt{n})$ |

a complicated way of defining a linear bandit. The name comes from the fact that in general elements of $\mathcal{E}$ are assumed to be low-rank. The precise nature of the problem is determined by assumptions on $\mathcal{E}$ and the action set $\mathcal{A}$. Our setup is recovered by assuming that $\mathcal{E} = \{xx^\top : x \in \mathbb{S}_r^{d-1}\}$ and $\mathcal{A} = \{xx^\top : x \in \mathbb{B}_1^d\}$.

Jun et al. [2019] assume that $\mathcal{E}$ consists of rank $p$ matrices and $\mathcal{A} = \{xy^\top : x \in \mathcal{X}, y \in \mathcal{Y}\}$ for some reasonably bounded sets $\mathcal{X} \subset \mathbb{R}^{d_1}$ and $\mathcal{Y} \subset \mathbb{R}^{d_2}$. They prove that the regret is bounded by $\tilde{O}((d_1 + d_2)^{3/2}\sqrt{pn})$. These results cannot be applied directly to the phase retrieval bandit because of the product assumption on the action set. Lu et al. [2021] retain the assumption that $\mathcal{E}$ consists of rank-$p$ matrices, but relax the product form of the action set (while also allowing for generalised linear models). Relying only on mild boundedness assumptions, they show that the regret can be bounded by $\tilde{O}((d_1 + d_2)^{3/2}\sqrt{pn})$. For the bandit phase retrieval problem, $d_1 = d_2 = d$ and $p = 1$, so this algorithm yields an upper bound on the regret for bandit phase retrieval of $\tilde{O}(d^{3/2}\sqrt{n})$. Both Jun et al. [2019] and Lu et al. [2021] conjecture that their upper bounds are optimal. Our results show that this is not true for this sub-problem, despite the fact that the heuristic argument used by these authors holds in this case, as we explain in Section 3. We summarize these comparisons in Table 1.

Some authors use a model where the noise is in the parameter rather than additive, which means the reward is $\langle A_t, \Theta_t \rangle$ with $(\Theta_t)_{t=1}^n$ an independent and identically distributed sequence of low-rank matrices (with unknown distribution). For example, Katariya et al. [2017b,a] and Trinh et al. [2020] assume that $\Theta_t$ is rank-1 almost surely and $\mathcal{A} = \{e_i e_j^\top : 1 \le i, j \le d\}$, which means the learner is trying to identify the largest entry in a matrix.

**Adversarial setting** A similar problem has been studied in the adversarial framework by Kotłowski and Neu [2019]. They assume that $(\theta_t)_{t=1}^n$ is a sequence of vectors chosen in secret by an adversary at the start of the game and the learner observes $\langle A_t, \theta_t \rangle^2$. They design an algorithm for which the regret is at most $\tilde{O}(d\sqrt{n})$, while the best lower bound is $\Omega(\sqrt{dn})$.

## 3 Information-theoretic heuristics and information-directed sampling

Jun et al. [2019, §5] argue by comparing the signal to noise ratios between linear and low-rank bandits that the minimax regret for problems like bandit phase retrieval should be lower bounded by $\Omega(d^{3/2}\sqrt{n})$. We make this argument a little more formal and explain why it does not yield the right answer in this instance. Let $\lambda$ be the uniform (rotational invariant) measure on $\mathbb{S}_r^{d-1}$ and suppose that $\theta_\star$ is sampled from $\lambda$. The learner takes an action $A \in \mathbb{B}_1^d$ and observes $X = \langle A, \theta_\star \rangle^2 + \eta$ with $\eta$ sampled from a standard Gaussian. What is the information gained by the learner? By symmetry, all actions on the unit sphere have the same information gain, so let's just fix an arbitrary $A \in \mathbb{S}_1^{d-1}$. Let

$\mathbb{P}_\theta$ be a Gaussian with mean $\langle A, \theta \rangle^2$ and variance 1. Then,

$$
\begin{aligned}
I(\theta_\star; X) &= \mathbb{E}\left[ \mathrm{KL}\left( \mathbb{P}_{\theta_\star}, \int_{\mathbb{S}_r^{d-1}} \mathbb{P}_\theta \, \mathrm{d}\lambda(\theta) \right) \right] = \int_{\mathbb{S}_r^{d-1}} \mathrm{KL}\left( \mathbb{P}_\alpha, \int_{\mathbb{S}_r^{d-1}} \mathbb{P}_\theta \, \mathrm{d}\lambda(\theta) \right) \mathrm{d}\lambda(\alpha) \\
&\leq \int_{\mathbb{S}_r^{d-1}} \int_{\mathbb{S}_r^{d-1}} \mathrm{KL}\left( \mathbb{P}_\alpha, \mathbb{P}_\theta \right) \mathrm{d}\lambda(\alpha) \, \mathrm{d}\lambda(\theta) = \frac{1}{2} \int_{\mathbb{S}_r^{d-1}} \int_{\mathbb{S}_r^{d-1}} \left( \langle A, \theta \rangle^2 - \langle A, \alpha \rangle^2 \right)^2 \mathrm{d}\lambda(\alpha) \, \mathrm{d}\lambda(\theta) \\
&= \frac{r^4}{2} \left( \frac{3}{d^2 + 2d} - \frac{1}{d^2} \right) \leq \frac{r^4}{d^2} \,,
\end{aligned}
\tag{1}
$$

where the first inequality follows from convexity of the relative entropy. Note that $\frac{d}{2} \log(n)$ bits are needed to code $\theta_\star$ to reasonable accuracy. So if we presume that the rate of information learned *stays the same* throughout the learning process, then over $n$ rounds the learner can only obtain $O(nr^4/d^2)$ bits by Eq. (1). By setting $r^2 = d^{3/2} \sqrt{\log(n)/n}$ one could be led to believe that the learner cannot identify the optimal direction and the regret would be $\Omega(nr^2) = \Omega(d^{3/2}\sqrt{n \log(n)})$. The main point is that the rate of information accumulated by a careful learner *increases* over time.

**Information-directed sampling** The combination of our upper bound and the observation above has an important implication. Suppose as above that $\theta_\star$ is sampled uniformly from $\mathbb{S}_r^{d-1}$ and let $A$ be a possibly randomised action. Since the learner cannot know the realisation of $\theta_\star$ initially, her expected regret for any action $a \in \mathbb{B}_1^d$ is $\Delta(a) = r^2 - \mathbb{E}[\langle a, \theta_\star \rangle^2] = r^2(1 - \|a\|^2/d) \geq r^2(1 - 1/d)$. On the other hand, as outlined above, the information gain about $\theta_\star$ is about $r^4/d^2$. Together these results show that the information ratio is bounded by

$$
\Psi \triangleq \frac{\mathbb{E}[\Delta(A)]^2}{I(\theta_\star; X, A)} = \Theta(d^2) \,.
$$

Since the entropy of a suitable approximation of the optimal action is about $\frac{d}{2} \log(n)$, an application of the information theoretic analysis by Russo and Van Roy [2014] suggests that the Bayesian regret can be bounded by $O(d^{3/2}\sqrt{n \log(n)})$, which is suboptimal. This time the problem is that we have used the worst-case bound on the information ratio, without taking into account the possibility that the information ratio might decrease over time. We should mention here that a decreasing information ratio was exploited by Devraj et al. [2021] in a recent analysis of Thompson sampling for finite-armed bandits, but there the gain was less dramatic (a logarithm of the number of arms) and no changes to the algorithm were required.

## 4   Algorithm for bandit phase retrieval

We start by showing that Theorem 1 holds if the learner is given an action that is constant-factor optimal. In the next section we explain how such an action can be identified with low regret. Our algorithm uses the explore-then-commit design principle, which is usually only sufficient for $O(n^{2/3})$ regret. The reason we are able to obtain $O(n^{1/2})$ regret is because of the curvature of the action set, a property that has been exploited in a similar way in a variety of settings [Rusmevichientong and Tsitsiklis, 2010a, Huang et al., 2017, Kirschner et al., 2020].

**Theorem 4.** *Suppose the learner is given an action $\widehat{A}_w \in \mathbb{B}_1^d$ such that $\langle \widehat{A}_w, \theta_\star \rangle^2 \geq \alpha r^2$ for some universal constant $\alpha \in (0, 1]$. Then there exists a policy $\pi$ for which the regret is at most $\mathfrak{R}_n(\pi, \theta_\star) \leq \mathrm{const} \cdot d\sqrt{n \log(n)}$.*

*Proof.* By choosing the sign of $\theta_\star$, assume without loss of generality that $\langle \widehat{A}_w, \theta_\star \rangle \geq r\sqrt{\alpha}$. Let

$$
m = \left\lceil 4d\sqrt{n \log(n)}/r^2 \right\rceil \qquad \text{and} \qquad \lambda = \min\left( \frac{1}{2}, \frac{\sqrt{\alpha}}{4} \right) \,.
$$

If $m \geq n$, then the regret of any policy is upper bounded $nr^2 \leq \mathrm{const}\, d\sqrt{n \log(n)}$, so for the rest of the proof we assume that $m < n$. For the first $m$ rounds the policy cycles over the $2d$ actions $\{(1-\lambda)\widehat{A}_w \pm \lambda e_k : k \in [d]\}$. The constrained least squares estimator of $\theta_\star$ based on the data collected over $m$ rounds is

$$
\hat{\theta} = \arg\min\{\mathcal{L}(\theta) : \theta \in \mathbb{B}_r^d \text{ and } \langle \widehat{A}_w, \theta \rangle \geq r\sqrt{\alpha}\} \,,
\tag{2}
$$

where $\mathcal{L}(\theta) = \frac{1}{2}\sum_{t=1}^{m}(X_t - \langle A_t, \theta\rangle^2)^2$. For the remaining $n - m$ rounds the algorithm plays $A_t = \widehat{A} = \hat\theta/\|\hat\theta\|$. Then,

$$\beta \triangleq 9 + d\log(98m) \tag{3}$$

$$\geq \mathbb{E}\left[\sum_{t=1}^{m}\langle A_t, \hat\theta - \theta_\star\rangle^2 \langle A_t, \hat\theta + \theta_\star\rangle^2\right] \qquad \text{(by Corollary 10)}$$

$$\geq \frac{\alpha\|\theta_\star\|^2}{4}\mathbb{E}\left[\sum_{t=1}^{m}\langle A_t, \hat\theta - \theta_\star\rangle^2\right] \tag{4}$$

$$\geq \frac{\alpha\lambda^2\|\theta_\star\|^2}{2}\left(\frac{m}{2d} - 1\right)\mathbb{E}\left[\|\hat\theta - \theta_\star\|^2\right]. \tag{5}$$

where in Eq. (4) we used that for some $k \in [d]$,

$$\langle A_t, \hat\theta + \theta_\star\rangle = \langle(1-\lambda)\widehat{A}_w \pm \lambda e_k, \hat\theta + \theta_\star\rangle \geq (1-\lambda)\langle\widehat{A}_w, \hat\theta + \theta_\star\rangle - \lambda\|\hat\theta + \theta_\star\|$$

$$\geq 2(1-\lambda)\sqrt{\alpha}r - 2\lambda r \geq \frac{\sqrt{\alpha}}{2}\|\theta_\star\|. \qquad \text{(by definition of } \lambda)$$

Eq. (5) follows because for any $k \in [d]$,

$$\sum_{\sigma \in \pm 1}\langle(1-\lambda)\widehat{A}_w + \sigma\lambda e_k, \hat\theta - \theta_\star\rangle^2 = 2(1-\lambda)^2\langle\widehat{A}_w, \hat\theta - \theta_\star\rangle^2 + 2\lambda^2\langle e_k, \hat\theta - \theta_\star\rangle^2$$

$$\geq 2\lambda^2\langle e_k, \hat\theta - \theta_\star\rangle^2,$$

which implies that

$$\sum_{t=1}^{m}\langle A_t, \hat\theta - \theta_\star\rangle^2 \geq \left\lfloor\frac{m}{2d}\right\rfloor\sum_{k=1}^{d}\left(2\lambda^2\langle e_k, \hat\theta - \theta_\star\rangle^2\right) \geq 2\lambda^2\left(\frac{m}{2d} - 1\right)\|\hat\theta - \theta_\star\|^2.$$

Rearranging Eq. (5) and using the definition of $\beta$ shows that

$$\mathbb{E}\left[\|\hat\theta - \theta_\star\|^2\right] \leq \frac{2\beta}{\alpha\lambda^2\|\theta_\star\|^2}\frac{1}{\frac{m}{2d} - 1} \leq \text{const}\frac{d^2\log(m)}{m\|\theta_\star\|^2}.$$

Letting $a_\star = \theta_\star/\|\theta_\star\|$ be the optimal action, the regret is bounded by

$$\mathfrak{R}_n(\pi, \theta_\star) \leq m\|\theta_\star\|^2 + n\mathbb{E}\left[\langle a_\star, \theta_\star\rangle^2 - \langle\widehat{A}, \theta_\star\rangle^2\right]$$

$$= m\|\theta_\star\|^2 + n\mathbb{E}\left[\langle a_\star - \widehat{A}, \theta_\star\rangle\langle a_\star + \widehat{A}, \theta_\star\rangle\right]$$

$$\leq m\|\theta_\star\|^2 + 2n\mathbb{E}\left[\left|\langle a_\star - \widehat{A}, \theta_\star\rangle\right|\|\theta_\star\|\right]$$

$$\leq m\|\theta_\star\|^2 + 4n\mathbb{E}\left[\|\theta_\star - \hat\theta\|^2\right] \qquad \text{(by Lemma 6)}$$

$$\leq m\|\theta_\star\|^2 + \text{const}\frac{nd^2\log(m)}{m\|\theta_\star\|^2} \leq \text{const}\,d\sqrt{n\log(n)}. \qquad \square$$

## 5  Finding a constant-factor optimal action

To establish Theorem 1 we show there exists an algorithm that interacts with the bandit for a random number of rounds and outputs an action $\widehat{A}_w$ that with high probability satisfies $\langle\widehat{A}_w, \theta_\star\rangle^2 \geq r^2/64$. Furthermore, the procedure suffers small regret in expectation.

**Theorem 5.** *Let $T$ be the random number of rounds that Algorithm 1 interacts with the bandit, which cannot be more than $n$, and let $\widehat{A}_w \in \mathbb{S}_1^{d-1}$ be its output. Then,*

1. *$\mathbb{E}[T] \leq \text{const}\frac{d^2}{r^4}\log(n)\log(d)$.*

2. *With probability at least $1 - 1/n$, either $T = n$ or $\langle\widehat{A}_w, \theta_\star\rangle^2 \geq \frac{r^2}{64}$.*

```
1   τ = r/√d  and  m = ⌈ 8/τ⁴ log(2n²) ⌉
2   do
3       sample v uniformly from 𝕊₁^{d-1}
4       play Aₜ = v for m rounds and compute average reward X̄
5   loop until X̄ ≥ τ²
6   ℰ = {v√X̄}
7   for k = 2 to d

8       β = 9(log(98) + 4log(n))  and  m = ⌈ 64d²β/kr⁴ ⌉  and  u = Σ_{w∈ℰ} w / ‖Σ_{w∈ℰ} w‖

9       do
10          sample v uniformly from span(ℰ)^⊥ ∩ 𝕊₁^{d-1}
11          play Aₜ ∈ {(u+v)/√2, (u-v)/√2, u} for 3m rounds
12          find least squares estimator θ̂ constrained to 𝔹₁^d
13      loop until ⟨v, θ̂⟩² ≥ τ²
14      ℰ = ℰ ∪ {⟨v, θ̂⟩v}
15      if Σ_{w∈ℰ} ‖w‖² ≥ r²/16 break
16  end for
17  return Â_w = Σ_{w∈ℰ} w / ‖Σ_{w∈ℰ} w‖
```

**Algorithm 1:** The procedure operates in $d$ iterations. The first iteration is implemented in Lines 1–5 and the remaining $d-1$ iterations in Lines 7–15.

What is interesting about Algorithm 1 is that it uses what it has learned in early iterations to increase the statistical efficiency of its estimation.

*Proof.* Note that the vectors $u$ and $v$ computed in each iteration are orthogonal, which means that $\|(u+v)/\sqrt{2}\| = \|(u-v)/\sqrt{2}\| = \|v\| = 1$. Hence the actions of the algorithm are always in $\mathbb{B}_1^d$. The main argument of the proof is based on an induction to show that with high probability when the execution of the algorithm ends, there exists an $s \in \{\pm 1\}$ such that for all $w \in \mathcal{E}$,

(a) $w \in \mathcal{E}$, $\|w\|^2 \geq \tau^2$; and

(b) $s\langle w, \theta_\star \rangle \in [\frac{1}{2}\|w\|^2, 2\|w\|^2]$.

We proceed in five steps. First, we prove that if the above holds and the algorithm halts before $n$ rounds are over, then the vector returned is a suitable approximation of $\theta_\star/\|\theta_\star\|$. Second, we upper bound the probability of certain bad events. In the third and fourth steps we prove the base case and induction step for (a) and (b). In the last step we bound the expected running time.

**Step 1: Correctness** Suppose that (a) and (b) above hold and the algorithm halts at the end of iteration $k$. Then,

$$\langle \widehat{A}_w, \theta_\star \rangle^2 = \left\langle \frac{\sum_{w\in\mathcal{E}} w}{\|\sum_{w\in\mathcal{E}} w\|}, \theta_\star \right\rangle^2 \geq \frac{1}{4} \sum_{w\in\mathcal{E}} \|w\|^2 \geq \frac{r^2}{64}.$$

where the first inequality follows from orthogonality of $w \in \mathcal{E}$ and (b) above. The second inequality follows from the stopping condition in Line 15 of Algorithm 1, (a) above and the definition of $\tau$. Part (2) of the theorem follows by showing that (a) and (b) above hold with probability at least $1 - 1/n$.

**Step 2: Failure events** The algorithm computes some kind of estimator at the end of each do/loop. Since the algorithm cannot play more than $n$ actions, the number of estimators computed is naively upper bounded by $n$. A union bound over all estimates and the concentration bounds in Lemma 7 and Corollary 10 show that with probability at least $1 - 1/n$ the following both hold:

- For all $v$ sampled in the first iteration and corresponding average rewards $\bar{X}$,

$$\left|\bar{X} - \langle v, \theta_\star \rangle^2\right| \leq \sqrt{\frac{2\log(2n^2)}{m}} = \frac{\tau^2}{2}, \tag{6}$$

where $m$ is defined in Line 1 of Algorithm 1.

- Let $\mathcal{D} = (A_s)_s$ be the actions played in the inner loop of some iteration $k \geq 2$ and $\hat{\theta}$ be the corresponding least-squares estimator. Then,

$$\sum_{a \in \mathcal{D}} \langle a, \hat{\theta} - \theta_\star \rangle^2 \langle a, \hat{\theta} + \theta_\star \rangle^2 \leq 9(\log(98) + 4\log(n)) \triangleq \beta. \tag{7}$$

We assume both of the above hold in all relevant iterations for the remainder.

**Step 3: Base case** The next step is to show that (a) and (b) hold with high probability after the first iteration. Consider the operation of the algorithm in the inner loop. After sampling $v \in \mathbb{S}_1^{d-1}$, the algorithm plays $v$ for $m$ rounds and computes the average reward. Let $v$ be the last sampled action before the iteration halts and $w = v\sqrt{\bar{X}}$. By the stopping condition in Line 5, $\|w\|^2 = \bar{X} \geq \tau^2$. Without loss of generality, we choose the sign of $\theta_\star$ so that $\langle v, \theta_\star \rangle \geq 0$. Then by Eq. (6),

$$\langle w, \theta_\star \rangle = \sqrt{\bar{X}} \langle v, \theta_\star \rangle \in \left[\frac{1}{2}\|w\|^2, 2\|w\|^2\right].$$

This establishes the base case.

**Step 4: Inductive step** Assume that (a) and (b) above hold for $\mathcal{E}$ at the end of iteration $k$. Let $u$ be the value computed in Line 8 of Algorithm 1. Then,

$$\langle u, \theta_\star \rangle = \frac{\sum_{w \in \mathcal{E}} \langle w, \theta_\star \rangle}{\sqrt{\sum_{w \in \mathcal{E}} \|w\|^2}} \geq \frac{1}{2}\sqrt{\sum_{w \in \mathcal{E}} \|w\|^2} \geq \frac{\tau\sqrt{k}}{2}.$$

Let $\mathcal{A} = \{(u+v)/\sqrt{2}, (u-v)/\sqrt{2}, v\}$, which is the set of actions played in the inner loop of iteration $k+1$ after sampling $v$ for the last time. Let $\hat{\theta}$ be the corresponding least-squares estimate. We consider two cases. First, if $\langle u, \hat{\theta} + \theta_\star \rangle \geq 2|\langle v, \hat{\theta} + \theta_\star \rangle|$, then by Eq. (7),

$$\begin{aligned}
\frac{\beta}{m} &\geq \sum_{a \in \mathcal{A}} \langle a, \theta_\star - \hat{\theta} \rangle^2 \langle a, \theta_\star + \hat{\theta} \rangle^2 \\
&\geq \frac{1}{16} \langle u+v, \theta_\star - \hat{\theta} \rangle^2 \langle u, \theta_\star \rangle^2 + \frac{1}{16} \langle u-v, \theta_\star - \hat{\theta} \rangle^2 \langle u, \theta_\star \rangle^2 \\
&\geq \frac{1}{8} \langle v, \theta_\star - \hat{\theta} \rangle^2 \langle u, \theta_\star \rangle^2 \geq \frac{kr^2}{16d} \langle v, \theta_\star - \hat{\theta} \rangle^2.
\end{aligned}$$

Rearranging shows that

$$\langle v, \theta_\star - \hat{\theta} \rangle^2 \leq \frac{16d\beta}{mkr^2} \leq \frac{\tau^2}{4}. \tag{8}$$

For the second case, $\langle u, \hat{\theta} + \theta_\star \rangle \leq 2|\langle v, \hat{\theta} + \theta_\star \rangle|$. Then,

$$\frac{\beta}{m} \geq \sum_{a \in \mathcal{A}} \langle a, \theta_\star - \hat{\theta} \rangle^2 \langle a, \theta_\star + \hat{\theta} \rangle^2 \geq \frac{1}{4} \langle v, \hat{\theta} - \hat{\theta} \rangle^2 \langle u, \hat{\theta} + \theta_\star \rangle \geq \frac{kr^2}{8d} \langle v, \theta_\star - \hat{\theta} \rangle^2.$$

And again, Eq. (8) holds. Summarising, $\langle v, \hat{\theta} \rangle$ is an estimator of $\langle v, \theta_\star \rangle$ up to accuracy $\tau/2$. By the definition of the algorithm, the iteration only ends if $|\langle v, \hat{\theta} \rangle| \geq \tau$. Therefore, with $w = \langle v, \hat{\theta} \rangle v$, we have $\|w\|^2 = \langle v, \hat{\theta} \rangle^2 \geq \tau^2$. Furthermore, $\langle w, \theta_\star \rangle = \langle v, \hat{\theta} \rangle \langle v, \theta_\star \rangle \in [\|w\|^2/2, 2\|w\|^2]$. Therefore if (a) and (b) hold for $\mathcal{E}$ computed after iteration $k$, they also hold for $\mathcal{E}$ computed after iteration $k+1$.

**Step 5: Running time** The length of an iteration is determined by the corresponding value of $m$ and the number of samples of $v$. The former is an iteration-dependent constant, while the latter depends principally on how many samples are needed before $|\langle v, \theta_\star \rangle|$ is suitably large. The law of $v$ is the

uniform distribution on $\mathbb{S}_1^{d-1} \cap \mathrm{span}(\mathcal{E})^\perp$, which is the uniform distribution on a sphere of dimension $d - 1 - |\mathcal{E}|$ embedded in $\mathbb{R}^d$. The squared norm of the projection of $\theta_\star$ onto $\mathrm{span}(\mathcal{E})^\perp$ is

$$\|\theta_\star\|^2 - \sum_{w \in \mathcal{E}} \frac{\langle \theta_\star, w \rangle^2}{\|w\|^2} \geq r^2 - 4 \sum_{w \in \mathcal{E}} \|w\|^2 \geq \frac{r^2}{2},$$

where we used (a) of the induction and the stopping condition in Line 15. Therefore, when $v$ is sampled uniformly from $\mathbb{S}_1^{d-1} \cap \mathrm{span}(\mathcal{E})^\perp$, by Lemma 8,

$$\mathbb{P}\left(\langle v, \theta_\star \rangle^2 \geq 3\tau^2/2\right) = \mathbb{P}\left(\langle v, \theta_\star \rangle^2 \geq \frac{3r^2}{2d}\right) \geq \mathrm{const} > 0,$$

Furthermore, by the concentration analysis in the previous step, an iteration will end once a $v$ has been sampled for which $\langle v, \theta_\star \rangle^2 \geq 3\tau^2/2$. Hence, the expected number of times the algorithm samples $v$ per iteration is constant and a simple calculation using the definition of $m$ in Lines 1 and 8 shows that the expected number of rounds used by the algorithm is at most

$$\mathbb{E}[T] \leq \mathrm{const}\, \frac{d^2}{r^4} \log(n) \log(d). \qquad \square$$

## 6 Proof of Theorem 1 and Theorem 2

*Proof of Theorem 1.* Run Algorithm 1 and if it halts, feed the returned action to the input of the explore-then-commit algorithm analysed in Theorem 4. Algorithm 1 fails to return a suitable action with probability at most $1/n$, so the contribution of this event to the regret is negligible. By Theorem 5, the regret incurred by Algorithm 1 is bounded by

$$\mathbb{E}\left[\sum_{t=1}^{T}\left(r^2 - \langle A_t, \theta_\star \rangle^2\right)\right] \leq r^2 \mathbb{E}[T]$$

$$\leq \min\left(nr^2, \mathrm{const}\, \frac{d^2}{r^4} \log(d) \log(n)\right)$$

$$\leq \mathrm{const}\, d\sqrt{n \log(d) \log(n)}.$$

Combining this with the regret bound established in Theorem 4 yields the result. $\qquad \square$

*Proof of Theorem 2.* We use a standard reduction [Lattimore and Szepesvári, 2020, Chapter 33]. Let $\pi$ be the policy used in the proof of Theorem 1 with $\widehat{A}_\star$ sampled uniformly from $(A_t)_{t=1}^n$. By Theorem 1,

$$\mathfrak{r}_n(\pi, \theta_\star) = \frac{1}{n}\left(nr^2 - \mathbb{E}\left[\sum_{t=1}^{n} \langle A_t, \theta_\star \rangle^2\right]\right) = \frac{\mathfrak{R}_n(\pi, \theta_\star)}{n} \leq \mathrm{const}\, d\sqrt{\frac{\log(n) \log(d)}{n}}.$$

$$\square$$

## 7 Proof of Theorem 3

Let $\pi$ be a fixed policy and for $\theta \in \mathbb{R}^d$ let $\mathbb{P}_\theta$ be the measure on the sequence of outcomes $H_n = (A_1, X_1, \ldots, A_n, X_n)$ induced by the interaction between $\pi$ and the phase retrieval model determined by $\theta$. Let $\mathbb{E}_\theta$ denote the expectation with respect to $\mathbb{P}_\theta$. Let $r$ be a positive constant to be tuned subsequently and $\sigma$ be the uniform (Haar) measure on $\mathbb{S}_r^{d-1}$. Let $\mathbb{Q} = \int \mathbb{P}_\theta \, \mathrm{d}\sigma(\theta)$ be the Bayesian mixture measure. For $\theta \in \mathbb{R}^d$, let $\mathcal{E}_\theta$ be the event given by

$$\mathcal{E}_\theta = \left\{ \langle \widehat{A}_\star, \theta \rangle^2 \geq \frac{3}{4} r^2 \right\}.$$

By Fano's inequality [Gerchinovitz et al., 2020, Lemma 5],

$$\int_{\mathbb{S}_r^{d-1}} \mathbb{P}_\theta(\mathcal{E}_\theta) \, \mathrm{d}\sigma(\theta) \leq \frac{\log 2 + \int_{\mathbb{S}_r^{d-1}} \mathrm{KL}(\mathbb{P}_\theta, \mathbb{Q}) \, \mathrm{d}\sigma(\theta)}{-\log\left(\int_{\mathbb{S}_r^{d-1}} \mathbb{Q}(\mathcal{E}_\theta) \, \mathrm{d}\sigma(\theta)\right)}. \tag{9}$$

We now bound the numerator and denominator in Eq. (9) to show that the right-hand side is at most $1/2$ and then complete the proof using the definition of the regret and $\mathcal{E}_\theta$.

**Step 1: Bounding the denominator in Eq. (9)** By exchanging the order of integrals in the denominator of Eq. (9), it follows that

$$- \log \left( \int_{\mathbb{S}_r^{d-1}} \mathbb{Q}(\mathcal{E}_\theta) \, d\sigma(\theta) \right) = - \log \left( \int \int_{\mathbb{S}_r^{d-1}} \mathbf{1}_{\mathcal{E}_\theta} \, d\sigma(\theta) \, d\mathbb{Q} \right) . \tag{10}$$

If $U$ is sampled uniformly from $\mathbb{S}_1^{d-1}$, then by a concentration bound for spherical measures [Dasgupta and Gupta, 2003, Lemma 2.2],

$$\mathbb{P} \left( U_1^2 \geq \delta/d \right) \leq \exp(-\delta/4) \text{ for all } \delta > 6 .$$

By scaling and rotating and choosing $\delta = \frac{3}{4}d$, it follows that for any $\widehat{A}_\star \in \mathbb{B}_1^d$,

$$\int_{\mathbb{S}_r^{d-1}} \mathbf{1} \left( \langle \widehat{A}_\star, \theta \rangle^2 \geq \frac{3r^2}{4} \right) d\sigma(\theta) \leq \exp\left( -3d/16 \right) .$$

Therefore, by Eq. (10),

$$- \log \left( \int_{\mathbb{S}_r^{d-1}} \mathbb{Q}(\mathcal{E}_\theta) \, d\sigma(\theta) \right) \geq \frac{3d}{16} .$$

**Step 2: Bounding the numerator in Eq. (9)** By the convexity of KL-divergence,

$$\int_{\mathbb{S}_r^{d-1}} \mathrm{KL}(\mathbb{P}_\theta, \mathbb{Q}) \, d\sigma(\theta) = \int_{\mathbb{S}_r^{d-1}} \mathrm{KL} \left( \mathbb{P}_\theta, \int_{\mathbb{S}_r^{d-1}} \mathbb{P}_\alpha \, d\sigma(\alpha) \right) d\sigma(\theta)$$

$$\leq \int_{\mathbb{S}_r^{d-1}} \int_{\mathbb{S}_r^{d-1}} \mathrm{KL} \left( \mathbb{P}_\theta, \mathbb{P}_\alpha \right) d\sigma(\alpha) \, d\sigma(\theta) .$$

By the chain rule of KL-divergence,

$$\mathrm{KL} \left( \mathbb{P}_\theta, \mathbb{P}_\alpha \right) = \mathbb{E}_\theta \left[ \sum_{t=1}^n \mathrm{KL} \left( \mathbb{P}_\theta(Y_t = \cdot | A_t), \mathbb{P}_\alpha(Y_t = \cdot | A_t) \right) \right] .$$

A straightforward computation leads to

$$\mathrm{KL} \left( \mathbb{P}_\theta, \mathbb{P}_\alpha \right) = \mathbb{E}_\theta \left[ \sum_{t=1}^n \frac{1}{2} \left( (A_t^\top \theta)^2 - (A_t^\top \alpha)^2 \right)^2 \right]$$

$$= \frac{1}{2} \mathbb{E}_\theta \left[ \sum_{t=1}^n \left( (A_t^\top \theta)^4 - 2(A_t^\top \theta)^2 (A_t^\top \alpha)^2 + (A_t^\top \alpha)^4 \right) \right] .$$

Since $(A_t)_{t=1}^n$ are independent of $(X_t)_{t=1}^n$, we can interchange the expectation and integral such that

$$\int_{\mathbb{S}_r^{d-1}} \mathrm{KL}(\mathbb{P}_\theta, \mathbb{Q}) \, d\sigma(\theta)$$

$$\leq \sum_{t=1}^n \mathbb{E} \left[ \int_\theta \int_\alpha \left( (A_t^\top \theta)^4 - 2(A_t^\top \theta)^2 (A_t^\top \alpha)^2 + (A_t^\top \alpha)^4 \right) d\sigma(\alpha) \, d\nu(\theta) \right] ,$$

where the expectation is with respect to $(A_t)_{t=1}^n$, which does not depend on $\theta$ by assumption. When $\theta$ is uniformly on $\mathbb{S}_r^{d-1}$ and $A \in \mathbb{B}_1^d$ is arbitrary,

$$\int_{\mathbb{S}_r^{d-1}} \langle A_t, \theta \rangle^4 \, d\sigma(\theta) = \frac{3r^4}{d^2 + 2d} \qquad \text{and} \qquad \int_{\mathbb{S}_r^{d-1}} \langle A_t, \theta \rangle^2 \, d\sigma(\theta) = \frac{1}{d^2} ,$$

where the expectation is taken with respect to $\theta$. Therefore,

$$\int_{\mathbb{S}_\theta^{d-1}} \mathrm{KL}(\mathbb{P}_\theta, \mathbb{Q}) \, d\sigma(\theta) \leq \frac{3nr^4}{d^2} . \tag{11}$$

**Step 3: Lower bounding the regret** Let $r^2 = \sqrt{d^3/(32n)}$ Combining the previous two steps shows that

$$\int_{\mathbb{S}_r^{d-1}} \mathbb{P}_\theta(\mathcal{E}_\theta) \, \mathrm{d}\sigma(\theta) \leq \frac{16nr^2}{d^3} \leq \frac{1}{2} \, .$$

Therefore there exists a $\theta \in \mathbb{S}_r^{d-1}$ with $\mathbb{P}_\theta(\mathcal{E}_\theta) \leq 1/2$, which implies that

$$\mathfrak{r}_n(\pi, \theta) = r^2 - \mathbb{E}_\theta \left[ \langle \widehat{A}_\star, \theta \rangle^2 \right] \geq \frac{r^2}{8} \geq \mathrm{const} \, \frac{d^{3/2}}{\sqrt{n}} \, .$$

## 8    Discussion

**Unknown radius** The assumption that $r = \|\theta_\star\|$ is known to the learner is easily relaxed by estimating $\|\theta_\star\|$. Note first that all our analysis holds with only trivial modifications if $r \in [\frac{1}{2}\|\theta_\star\|, \|\theta_\star\|]$. Next, if $A$ is sampled uniformly from $\mathbb{S}_1^{d-1}$ and $X = \langle A, \theta_\star \rangle^2 + \eta$ and $\eta$ is a standard Gaussian, then $\mathbb{E}[X] = \frac{1}{d}\|\theta_\star\|^2$ and $\mathbb{V}[X] = 1 + 2(d-1)/(d^3 + 2d^2) = \Theta(1)$. Therefore $\|\theta_\star\|$ can be estimated to within an arbitrary multiplicative factor and at confidence level $1 - 1/n$ using $\mathrm{const} \, d^2/\|\theta_\star\|^4 \log(n)$ interactions with the bandit.

**Computation complexity** The only computational challenge is finding the least squares estimates, which is a non-convex optimisation problem. Candès et al. [2015b] proposed a Wirtinger flow algorithm that starts with a spectral initialization, and then refines this initial estimate using a local update like gradient descent. The computational complexity of the Wirtinger flow algorithm with $\varepsilon$-accuracy is $O(nd^2 \log(1/\varepsilon))$ where $n$ is the number of samples.

**Adversarial setting** Kotłowski and Neu [2019] study the adversarial version of this problem, where the learner observes $\langle A_t, \theta_t \rangle^2$ and $(\theta_t)_{t=1}^n$ is an adversarially sequence with $\theta_t \in \mathbb{B}_1^d$ for all $t$. They prove an upper bound of $\mathfrak{R}_n = O(d\sqrt{n \log(n)})$ and a lower bound of $\Omega(\sqrt{dn})$. Natural attempts at improving the lower bound all fail. We believe that the upper bound is loose, but proving this remains delicate. No warm starting procedure will work anymore because the information gained may be useless in the presence of a change point. New ideas are needed.

**Rank-$p$** Perhaps the most natural open question is whether or not our analysis can be extended to the low rank bandit problem without our particular assumptions on the action set and environments matrices.

**Principled algorithms** Can optimism or information-directed sampling be made to work? The main challenge is to understand the sample paths of these algoritms *before* learning takes place.

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
