## A Technical lemma

**Lemma 6** (Kirschner et al. 2020). $\left\langle \frac{\theta}{\|\theta\|} - \frac{\varphi}{\|\varphi\|}, \theta \right\rangle \leq \frac{2}{\|\theta\|} \|\theta - \varphi\|^2.$

**Lemma 7** (Boucheron et al. 2013). *Let $(X_t)_{t=1}^n$ be independent standard Gaussian random variables and $(a_t)_{t=1}^n$ be constants. Then,*

$$\mathbb{P}\left( \left| \frac{1}{n} \sum_{t=1}^n a_t X_t \right| \geq \sqrt{\frac{2 \sum_{t=1}^n a_t^2 \log(2/\delta)}{n}} \right) \leq \delta.$$

**Lemma 8.** *Let $V \subset \mathbb{R}^d$ be a $m$-dimensional subspace and let $X$ be sampled uniformly from $\mathbb{S}_1^{d-1} \cap V$. Then for all $\varphi \in V$,*

$$\mathbb{P}\left( \langle X, \varphi \rangle^2 \geq \frac{\|\varphi\|^2}{m} \right) \geq \mathrm{const} > 0.$$

*Proof.* Use the fact that if $Z \in \mathbb{R}^m$ is a standard Gaussian, then

$$\langle X, \varphi \rangle^2 \stackrel{d}{=} \frac{Z_1 \|\varphi\|^2}{\|Z\|}.$$

Then use standard concentration for the Gaussian and $\chi$-squared distributions and naive union bounding [Laurent and Massart, 2000]. Alternatively, use the explicit form for the distribution of $X$ in combination with elementary bounds on the regularised incomplete beta function. □

## B Ordinary least squares

Here we provide some routine results for least-squares estimation of $\theta_\star$. Suppose that $(A_t)_{t=1}^n$ are fixed and $(\eta_t)_{t=1}^n$ are independent 1-subgaussian random variables and $X_t = \langle A_t, \theta_\star \rangle^2 + \eta_t$. The least-squares estimator of $\theta_\star$ constrained to $\Theta \subset \mathbb{B}_r^d$ is

$$\hat{\theta} = \underset{\theta \in \mathbb{B}^d}{\arg \min}\, \mathcal{L}(\theta) \qquad \text{with} \qquad \mathcal{L}(\theta) = \frac{1}{2} \sum_{t=1}^n \left( X_t - \langle A_t, \theta \rangle^2 \right)^2.$$

The symmetry of the problem means that $\mathcal{L}(\theta) = \mathcal{L}(-\theta)$ for all $\theta \in \mathbb{R}^d$, which means there is no hope that $\hat{\theta}$ might be close to $\theta_\star$. What is true is that for suitably exploratory $(A_t)$, $\hat{\theta}$ is close to either $\theta_\star$ or $-\theta_\star$.

**Theorem 9.** *Suppose that $\theta_\star \in \mathbb{B}_r^d$ and $\hat{\theta} = \arg\min_{\theta \in \Theta} \mathcal{L}(\theta)$. Then, for any $\delta \in (0, 1)$, with probability at least $1 - \delta$,*

$$\mathbb{P}\left( \sum_{t=1}^n \langle \hat{\theta} - \theta_\star, A_t \rangle^2 \langle \hat{\theta} + \theta_\star, A_t \rangle^2 \geq 9 \log\left( \frac{N_{1/(32n)}(\Theta)}{\delta} \right) \right) \leq \delta,$$

*where $N_\epsilon(\Theta) = \min\{|\mathcal{C}| : \mathcal{C} \subset \mathbb{R}^d, \forall x \in \Theta, \min_{y \in \mathcal{C}} \|x - y\| \leq \epsilon\}$.*

*Proof.* Since $\theta_\star \in \mathbb{B}^d$ by assumption, it follows that

$$0 \leq \mathcal{L}(\theta_\star) - \mathcal{L}(\hat{\theta})$$
$$= -\frac{1}{2} \sum_{t=1}^n \langle A_t, \hat{\theta} - \theta_\star \rangle^2 \langle A_t, \hat{\theta} + \theta_\star \rangle^2 + \sum_{t=1}^n \eta_t \langle A_t, \hat{\theta} - \theta_\star \rangle \langle A_t, \hat{\theta} + \theta_\star \rangle.$$

Let $\epsilon = 1/(32n)$ and $\mathcal{C} \subset \mathbb{R}^d$ be such that for all $x \in \Theta$ there exists a $y \in \mathcal{C}$ such that $\|x - y\| \leq \epsilon$ and $|\mathcal{C}| = N_\epsilon(\Theta)$. Since $A_t$ are fixed, by a union bound and standard Gaussian tail bounds, with probability at least $1 - |\mathcal{C}|\delta$,

$$\left| \sum_{t=1}^n \eta_t \langle A_t, \alpha - \theta_\star \rangle \langle A_t, \alpha + \theta_\star \rangle \right| \leq \sqrt{2 \sum_{t=1}^n \langle A_t, \alpha - \theta_\star \rangle^2 \langle A_t, \alpha + \theta_\star \rangle^2 \log\left( \frac{1}{\delta} \right)}.$$

On this event and letting $\alpha \in \mathcal{C}$ be such that $\|\alpha - \hat{\theta}\| \leq \epsilon$. Then, with $\Delta = \alpha - \hat{\theta}$,

$$\sum_{t=1}^{n} \langle A_t, \hat{\theta} - \theta_\star \rangle^2 \langle A_t, \hat{\theta} + \theta_\star \rangle^2 \leq \sqrt{8 \sum_{t=1}^{n} \langle A_t, \alpha - \theta_\star \rangle^2 \langle A_t, \alpha + \theta_\star \rangle^2 \log\left(\frac{1}{\delta}\right)}$$

$$= \sqrt{8 \sum_{t=1}^{n} \langle A_t, \Delta + \hat{\theta} - \theta_\star \rangle^2 \langle A_t, \Delta + \hat{\theta} + \theta_\star \rangle^2 \log\left(\frac{1}{\delta}\right)}$$

$$\leq \sqrt{8 \sum_{t=1}^{n} \left( \langle A_t, \hat{\theta} - \theta_\star \rangle^2 + 2\epsilon + \epsilon^2 \right) \left( \langle A_t, \hat{\theta} + \theta_\star \rangle^2 + 4\epsilon + \epsilon^2 \right) \log\left(\frac{1}{\delta}\right)}$$

$$\leq \sqrt{8 \sum_{t=1}^{n} \left( \langle A_t, \hat{\theta} - \theta_\star \rangle^2 \langle A_t, \hat{\theta} + \theta_\star \rangle^2 + 12\epsilon + 13\epsilon^2 + 6\epsilon^3 + \epsilon^4 \right) \log\left(\frac{1}{\delta}\right)}$$

$$\leq \sqrt{8 \sum_{t=1}^{n} \left( \langle A_t, \hat{\theta} - \theta_\star \rangle^2 \langle A_t, \hat{\theta} + \theta_\star \rangle^2 + 32\epsilon \right) \log\left(\frac{1}{\delta}\right)}$$

$$\leq \sqrt{\left( 1 + 8 \sum_{t=1}^{n} \langle A_t, \hat{\theta} - \theta_\star \rangle^2 \langle A_t, \hat{\theta} + \theta_\star \rangle^2 \right) \log\left(\frac{1}{\delta}\right)},$$

where in the final inequality we chose $\epsilon = 1/(32n)$. Solving for the left-hand side and naive simplification shows that

$$\sum_{t=1}^{n} \langle A_t, \hat{\theta} - \theta_\star \rangle^2 \leq 9 \log\left(\frac{1}{\delta}\right).$$

To summarise we have shown that with probability at least $1 - \delta$,

$$\sum_{t=1}^{n} \langle A_t, \hat{\theta} - \theta_\star \rangle^2 \langle A_t, \hat{\theta} + \theta_\star \rangle^2 \leq 9 \log\left(\frac{|\mathcal{C}|}{\delta}\right) = 9 \log\left(\frac{N_\epsilon(\Theta)}{\delta}\right). \qquad \square$$

Standard results show that when $\Theta \subset \mathbb{B}_1^d$ has dimension $k$, then $\log N_\epsilon(\Theta) \leq m \log(3/\epsilon)$. From this one obtains the following corollary:

**Corollary 10.** *Under the same conditions as Theorem 9 and when $\Theta \subset \mathbb{B}_1^d$ and $\dim(\text{span}(A_1, \ldots, A_n)) = k$:*

*(a)* $\mathbb{P}\left( \sum_{t=1}^{n} \langle A_t, \hat{\theta} - \theta_\star \rangle^2 \langle A_t, \hat{\theta} + \theta_\star \rangle^2 \geq 9 \left( \log(1/\delta) + k \log(98n) \right) \right) \leq \delta.$

*(b)* $\mathbb{E}\left[ \sum_{t=1}^{n} \langle A_t, \hat{\theta} - \theta_\star \rangle^2 \langle A_t, \hat{\theta} + \theta_\star \rangle^2 \right] \leq 9 \left( 1 + k \log(98n) \right).$