# OpenReview forum: "Bandit Phase Retrieval"
_NeurIPS.cc/2021/Conference — NeurIPS 2021 Poster_

### Official Review · Reviewer_A8GD · 2021-07-16

**Rating:** 7
**Confidence:** 3

**Summary:**

This paper studied the bandit phase retrieval problem that can also be framed as a low-rank bandit problem where the actions and the unknown coefficient are rank-1 matrices. Their algorithm achieves a \tilde{O}(d\sqrt{n}) regret that matches their lower bound result up to log factors. By a simple conversion, they prove a \tilde{O}(d/\sqrt{n}) simple regret and show this can only be achieved by adaptive algorithms.

**Limitations And Societal Impact:**

Yes.

**Main Review:**

This paper has strong theoretical results.
For low rank bandit problems (rank-1 action space and rank-1 unknown coefficient), they first develop an algorithm achieving \sqrt{T} regret that is only linear in d. Comparing to the results in Jun et. al and Lu et. al, their regret improves by a factor of \sqrt{d}. Their approach is also novel. But I don't think the author can claim the conjectured lower bounds in Jun et. al and Lu et. al are not true based on their \tilde{O}(d\sqrt{n}) regret upper bound. Because in both works of Jun et. al and Lu et. al, actions either have higher ranks or the action space doesn't have to be an exact Euclidean ball, which means their settings are more general and may have larger lower bounds.

I suggest the author re-write Section 3, include explanations for your arguments. Currently it's not easy to understand for people outside of non-information theory.

Minor comments:
1. equation below line 25: should be nr^2, not r^2
2. line 72: xx^T
3. equation below line 100: I(\theta_*; X)

Overall I think the theoretical contribution is strong and I recommend an acceptance.


**Time Spent Reviewing:**

4

---

> ### Author Response · Authors · 2021-08-07
> **Review Response**
>
> Thanks for the thoughtful review.
> 1. Regarding the conjectured lower bounds, we did not intend to claim they are wrong with certainty, only that our arguments and analysis point towards the possibility that they are not. While previously most experts would probably guess the upper bounds are tight, now we imagine there might be considerable uncertainty. We will rephrase these comments to make our thoughts clear.
> 2. Thanks for your suggestion about Section 3. We adapt Section 3 to make it approachable for the largest possible audience.
> 3. Thanks also for the minor comments. We will revise it accordingly.

---

### Official Review · Reviewer_fRir · 2021-07-16

**Rating:** 7
**Confidence:** 4

**Summary:**


The paper studies the quadratic bandit problem, where the reward at time $t$ is given by $\langle \theta_\star, a_t \rangle^2 + \varepsilon_t$, where $\varepsilon_t$ is i.i.d. gaussian noise, $\theta_\star$ is an unknown $d$-dimensional vector and $a$ is the action. The action set is fixed to be the unit ball, and the learner can choose any action $a_t \in \mathbb R^d$ such that $\lVert a_t \rVert_2 \le 1$. The norm of the unknown vector $\lVert \theta_\star \rVert$ is revealed to the learner at the beginning of the bandit. (This assumption can be removed, as discussed in the final section.) The main contributions are as follows.

1. The paper proves tight up to log factors upper and lower regret bounds $\tilde{\Theta}(d \sqrt n)$ for the quadratic bandit problem (Theorems 1, 2, respectively).
2. The paper extends the regret results to the pure exploration settings (Theorem 3).
3. The paper suggests that optimal regret is not attainable by fixed policies (Theorem 4).

The paper concludes that standard algorithms in bandits literature, including UCB, Thompson sampling, and information-directed sampling, might be insufficient for the quadratic bandit. Furthermore, the paper refutes previous regret lower bound conjecture, suggesting that naive guess based on signal-to-noise (SNR) ratio is wrong.

**Limitations And Societal Impact:**

### Limitations

1. The recent paper [1] studies the more general tensor bandit, where the reward function can be more general low-rank p-tensors. In particular, the $p=2$ case in [1] seems cover the main contributions (Theorem 1-3) in this paper. Besides, Algs. 1 and 2 in [1] look quite similar to Alg. 1 in the submission, but the ones in [1] are much more general. For example, the ones in [1] can solve general $\langle \theta_\star, a_t \rangle^p$ reward, while the submission only studies $\langle \theta_\star, a_t \rangle^2$. The authors may need to compare their results with those in [1].
2. The lower bound (Theorem 2) seems to be the same as [2, Theorem 6] by putting $r=1$. The authors might need to point it out more explicitly, and it would be strange to list it as a contribution.

### Other Comments

1. Line 72, what is $xx$?
2. At Line 125, the idea of using curvature to obtain sqrt regret for ETC could date back to even earlier papers, such as [3].


### Societal Impact

The paper is mainly theoretical, and I don't see any potential negative societal impact.

### Bibliography

[1] Huang, Baihe, Kaixuan Huang, Sham M. Kakade, Jason D. Lee, Qi Lei, Runzhe Wang, and Jiaqi Yang. "Optimal Gradient-based Algorithms for Non-concave Bandit Optimization." arXiv preprint arXiv:2107.04518 (2021).

[2] Lu, Yangyi, Amirhossein Meisami, and Ambuj Tewari. "Low-rank generalized linear bandit problems." In International Conference on Artificial Intelligence and Statistics, pp. 460-468. PMLR, 2021.

[3] Rusmevichientong, Paat, and John N. Tsitsiklis. "Linearly parameterized bandits." Mathematics of Operations Research 35, no. 2 (2010): 395-411.


**Main Review:**


Bandit with non-linear rewards is attracting increasing interests from the research society of bandits and RL, partly due to its close connection to deep RL. However, current research in this direction is quite limited. This paper studies the simplest task, namely the quadratic bandit, and gives near-optimal regret analyses. The paper conveys the important message that standard optimistic algorithms might be insufficient for quadratic bandit, and discovers the unusual fact that the SNR ratio might vary, and successfully exploits this fact to derive a near-optimal algorithm.

The paper designs a novel algorithm, which is a combination of a novel initialization stage (Alg. 1) ahead of the classical Explore-Then-Commit algorithm, to achieve optimal regret. The paper is well-written and the proofs are easy to follow. The authors are honest about their weakness about their results in the final section. However, I have some concerns on the paper. First, the related works might have been cited adequately, but the authors could have done better comparing with them. Second, there is a concurrent paper that covers many of the results in this paper. I vote for a weak reject.

---

The concurrent paper shall not be a concern. After discussion I would raise my vote to accept.

**Time Spent Reviewing:**

5

---

> ### Author Response · Authors · 2021-08-07
> **Review Response**
>
> Thanks for your careful review and the references. We would like to respond to your comments point by point.
> 1. Thanks for pointing out the nice recent paper by Huang et al..
>
> Note that this paper was uploaded to Arxiv on July 9, 2021 while the NeurIPS submission deadline was May 28, 2021. Of course we will include a detailed comparison with this work in subsequent revisions. While their manuscript covers more settings, our work is considerably shorter and we focus on a particular story with broad interest. At the same time, the analysis and algorithms are different in various ways, though both exploit the same phenomenon (as you would expect). We think there is value in accepting both papers.
>
> 2. Comparison with Theorem 6 in [2].
>
> Note that Theorem 6 in [2] can not be used as a lower bound in phase retrieval model since the true matrix parameter constructed in their proof is not positive semi-definite. But see our response to reviewer “gkvs”.
>
> 3. Line 72 should be $xx^{\top}$ and thanks for pointing out [3].

---

### Official Review · Reviewer_gkvs · 2021-07-16

**Rating:** 6
**Confidence:** 2

**Summary:**

The paper studies a stochastic bandit optimization problem in which the expected reward can be expressed as $\langle a , \theta_\star \rangle^2$,
where $\theta_\star$ is an unknown parameter.
The authors provides an algorithm that achieve a nearly tight (up to logarithmic factors) cumulative regret bounds.
Nearly tight bounds for simple regret have been shown as well.

**Limitations And Societal Impact:**

I think limitations and societal impact have been well discussed.

**Main Review:**

Originality:

The problem formulation is not very original because it is a special case of existing study,
e.g.,
linear bandits [APS11] and Bilinear bandits [JWWN19].
The proposed algorithm is based on the well-known explore-then-commit framework.
On the other hand,
the analysis contains non-trivial arguments in contrast to the standard analysis in the explore-then-commit framework,
which play an important role in achieving $O(\sqrt{n})$-regret.

I am a little skeptical that the lower bound (Theorem 2) is really a new result
as the proof of Theorem 8 in [KN19] provides $\Omega(d \sqrt{n / \log n})$ lower bounds
with stochastic losses expressed by a rank-1 matrix plus a noise term.
I guess that the models w.r.t.~noise are slightly different.
I would appreciate it if the authors could explain if there is an intrinsic difference.



Quality:

As far as I can see, no technical errors have been found.
I think the authors are careful and honest about evaluating both the strengths and weaknesses of their work.

Clarity:

The main part of the paper is clearly written.
One thing that was not clear to me was the relationship between this study and the existing studies [KN19] on the adversarial model.
Can the former be considered a special case of the latter?

The proof of Lemma 7 could not be found in the cited literature.
It would be nice to have more explicit pointers (I don't doubt the correctness of the claim).

Significance:

As the authors argue in Section 3, an analogy based on some existing studies suggests that the tight bound would be $\sim O(d^{3/2} \sqrt{n})$.
Therefore,
I think that the proved bound of $\tilde{\Theta}( d\sqrt{n} )$ is surprising and has a certain impact on the research community on bandit algorithms.


**Time Spent Reviewing:**

5

---

> ### Author Response · Authors · 2021-08-07
> **Review Response**
>
> Thanks for your careful and useful review. We would like to respond to your comments point by point.
> 1. About the lower bound in KN19.
>
> This is a great point! Indeed, although this lower bound was originally designed for rank-d adversarial problems, the rank-d part of the loss matrix in the proof has zero mean and the proof works in our setting. Obviously we’ll acknowledge this and probably either drop our proof, or push the analysis to the appendix if it seems like our novel proof can be valuable for others.
>
> 2. Adversarial model.
>
> In general the relationship between stochastic and adversarial models is a bit complicated because of the different noise models. For example, the KN19 analysis gives a $d \sqrt{n}$ upper bound for the rank-1 adversarial case, but the analysis fails in the stochastic setting (or would naively yield a $d^{3/2} \sqrt{n}$ bound. Their lower bound holds only for the rank-d case. The best known bound for the adversarial rank-1 case is $\sqrt{dn}$ and we guess this might be optimal. A similar and completely understood phenomenon appears in the linear bandit where the minimax adversarial regret on the sphere is $\sqrt{dn}$ while in the stochastic setting it is $d \sqrt{n}$. This difference is caused by the typical assumptions in the adversarial setting that the adversary plays in a set that guarantees the losses are bounded in $[0,1]$, which prevents the adversary from simulating (sub)Gaussian noise.
>
> 3. Proof of Lemma 7.
>
> Thanks for pointing this out. It comes from Lemma 19 in Kirschner et al. 2020. We will make this clear in the final version.

---

### Official Review · Reviewer_3qPJ · 2021-07-21

**Rating:** 7
**Confidence:** 4

**Summary:**

This paper studies the phase retrieval problem under bandit feedback. Regret bounds for this problem can be obtained from prior work by casting the problem as a low-rank bandit, which yields a $\tilde{O}(d^{3/2} \sqrt{n})$ regret upper bound. While several authors have conjectured that the upper bound for low-rank bandits is tight using information-theoretic heuristics, and these heuristics carry over also to the phase retrieval setting, this paper shows that these heuristics can be quite misleading. Indeed, this paper proves that the optimal regret for this setting is $\Theta(d \sqrt{n})$ (providing matching upper and lower bounds on the regret), which they obtain by ensuring that their algorithm can _increase_ the rate of information accumulated over time (thus circumventing the information-theoretic heuristics of prior works).


**Limitations And Societal Impact:**

Yes, adequately addressed.


**Main Review:**

Overall, I quite enjoyed both the results and the presentation of this paper.  The authors spend a significant portion of the paper devoted to discussing what is known about the bandit phase retrieval problem from prior work, and even sketch some heuristic arguments that were present in prior works, and why these arguments fail to give the correct answer in this setting. I found this discussion very enjoyable. The discussion here seems quite related to the separation in sample complexity between adaptive and non-adaptive algorithms for sparse recovery (see, e.g., “On the power of adaptivity in sparse recovery” by Indyk-Price-Woodruff in FOCS'11). It seems to me (at least at a high level), the reasons that the information-theoretic arguments for the phase retrieval setting fail are effectively the same reason why adaptive algorithms perform better in the sparse recovery setting. Perhaps this could be an additional discussion point in this section.

Regarding the analysis of the algorithm, I found the overall organization and flow to be relatively easy to follow. However, I think that adding some text explaining some intuition behind some of the steps involved in the proofs would improve the readability. For instance, (i) in the text preceding Theorem 5 (l.  124-125), the authors mention that they obtain $O(\sqrt{n})$ instead of $O(n^{2/3})$ scaling due to the curvature of the action set. As I understand it, I believe this is essentially lines 139-141, but it would be quite useful if the authors could highlight this in their analysis, and perhaps mention what would happen if the action set did not have the nice structure.  Additionally, (ii) a point that I found slightly subtle in the analysis, and perhaps could be useful to clarify, is that the result of Theorem 5 needs to hold _conditioned_ on the output of Algorithm 1 returning a good solution. This conditioning does not affect the application of Corollary 11 in the proof of Theorem 5, since the bandit feedback ensures the noise at each round is independent of the past. I suppose this is also a reason not to expect these ideas to carry over easily to the adversarial setting.

In summary, I think this paper would be a very nice addition to this year's conference. The ideas are interesting and somewhat surprising, the analysis is fairly simple, and the results are presented, for the most part, quite well.


**Time Spent Reviewing:**

7

---

> ### Author Response · Authors · 2021-08-07
> **Review Response**
>
> Thanks for your thoughtful review.
> 1. Nice observation about the prior work on sparse recovery. We will include a discussion in the final version.
> 2. About the curvature. We will highlight the curvature is used in Line 142 through Lemma 7.
> 3. Yes, you are right. Theorem 5 holds for any fixed $\hat{A}_w$. And in the proof of Theorem 1 in Line 207, we mention that Algorithm 1 fails to return suitable action with probability at most $1/n$, so the contribution of this event to the regret is negligible.

---

### Decision · Program_Chairs · 2021-09-27

**Decision:**

Accept (Poster)

**Comment:**

The paper studies the bandit phase retrieval problem and establishes a sharp minimax regret bound of d\sqrt{n}, which improves on previous  work based on low rank bandit approaches. The proof is enlightening and the paper is well-written, making nice connections to related work. All reviewers were positive on the paper, and so we advocate acceptance.

As I mentioned in our discussion, I do think it would be worthwhile to add some discussion of the recent/concurrent paper of Huang et al to the final manuscript.